# Functional and Genetic Diversity of Bacteria Associated with the Surfaces of Agronomic Plants

**DOI:** 10.3390/plants8040091

**Published:** 2019-04-04

**Authors:** Basharat Ali

**Affiliations:** Department of Microbiology and Molecular Genetics, University of the Punjab, Lahore 54590, Pakistan; basharat.ali.mmg@pu.edu.pk; Tel.: +92-300-4297937

**Keywords:** food biosafety, bacterial colonization, antibiotic resistance, biofilm formation, raw-eaten vegetables, bacterial auxin production

## Abstract

The main objective of this study was to evaluate the genetic diversity and agricultural significance of bacterial communities associated with the surfaces of selected agronomic plants (carrot, cabbage and turnip). The bacterial diversity of fresh agricultural produce was targeted to identify beneficial plant microflora or opportunistic human pathogens that may be associated with the surfaces of plants. Bacterial strains were screened in vitro for auxin production, biofilm formation and antibiotic resistance. 16S rRNA gene sequencing confirmed the presence of several bacterial genera including *Citrobacter*, *Pseudomonas*, *Pantoea*, *Bacillus*, *Kluyvera*, *Lysinibacillus*, *Acinetobacter*, *Enterobacter*, *Serratia*, *Staphylococcus*, *Burkholderia*, *Exiguobacterium*, *Stenotrophomonas*, *Arthrobacter* and *Klebsiella*. To address the biosafety issue, the antibiotic susceptibility pattern of strains was determined against different antibiotics. The majority of the strains were resistant to amoxicillin (25 µg) and nalidixic acid (30 µg). Strains were also screened for plant growth-promoting attributes to evaluate their positive interaction with colonized plants. Maximum auxin production was observed with *Stenotrophomonas maltophilia* MCt-1 (101 µg mL^−1^) and *Bacillus cereus* PCt-1 (97 µg mL^−1^). *Arthrobacter nicotianae* Lb-41 and *Exiguobacterium mexicanum* MCb-4 were strong biofilm producers. In conclusion, surfaces of raw vegetables were inhabited by different bacterial genera. Potential human pathogens such as *Bacillus cereus, Bacillus anthracis, Enterobacter cloacae, Enterobacter amnigenus* and *Klebsiella pneumoniae* were also isolated, which makes the biosafety of these vegetable a great concern for the local community. Nevertheless, these microbes also harbor beneficial plant growth-promoting traits that indicated their positive interaction with their host plants. In particular, bacterial auxin production may facilitate the growth of agronomic plants under natural conditions. Moreover, biofilm formation may help bacteria to colonize plant surfaces to show positive interactions with host plants.

## 1. Introduction

Fresh vegetables and fruits are integral components of the human diet and consumed in sufficient amounts to maintain good health [1]. There has been a correlation between foodborne outbreaks and increased production, imports and the consumption of fresh agricultural produce [2]. Raw vegetables can harbor pathogenic microorganisms that can cause serious illnesses to humans after consumption. Routes of microbial contamination include the application of organic wastes to soil as fertilizers, use of fecal-contaminated water for irrigation, direct contact with livestock and post-harvest hygiene issues [3]. In most common practices, fresh fruits and vegetables are consumed raw or processed to minimize the chances of foodborne outbreaks. The frequency of foodborne outbreaks from the consumption of contaminated vegetables and fruits has increased in recent decades. There is a long list of reports that confirms that raw vegetables foster several potentially pathogenic organisms, indicating their key involvement in fresh produce-associated outbreaks [4,5,6,7,8,9]. 

True or opportunistic human pathogens have been found and reported with foodborne outbreaks due to their adaptation to soil and plant surfaces [10]. They may be highly competitive for nutrients and produce antimicrobial agents to suppress the native microflora to colonize plant surfaces. For instance, pathogenic strains of *Pseudomonas aeruginosa* and *Stenotrophomonas maltophila* are well documented to colonize wheat and strawberries, respectively [11]. A similar ability of *Burkholderia cepacia* to cause infections in plants and humans has also been reported [12]. Besides the true human pathogens (*Escherichia coli* and *Salmonella enterica*), a number of potential human pathogens such as *Achromobacter xylosoxidans*, *Janthinobacterium lividum*, *Alcaligenes xylosoxidans*, *Alcaligenes faecalis*, *Serratia marcescens*, *Staphylococcus aureus*, *Enterobacter amnigenus*, *Bacillus cereus*, and *Stenotrophomonas maltophilia* have been reported to be present in plant-associated environments [13,14]. 

Plant-associated microorganisms can conveniently be categorized in three groups: plant-beneficial, plant pathogenic and opportunistic human pathogenic organisms. The plant-beneficial group of bacteria comprises members of different genera such as *Burkholderia*, *Enterobacter*, *Stenotrophomonas*, *Pseudomonas*, *Herbaspirillum*, *Ralstonia*, *Bradyrhizobium, Ochrobactrum, Bacillus, Rhizobium* and *Staphylococcus*. These microorganisms are mostly root-associated and factor in the bivalent interactions with plants and human or animal hosts [15]. The microorganisms of these genera proved themselves beneficial for plants by playing their role in growth promotion, protection from phytopathogens and provision of different elemental nutrients such as nitrogen and phosphorus [16,17]. Many potential pathogens of humans live in close association with plants and, therefore, have adapted to benefit plants in several ways in order to ensure their persistence. They can produce different vital phytohormones such as auxin, gibberellins and cytokinins. Some potentially pathogenic strains of *Bacillus* and *Serratia* are noted in the growth promotion of diverse crops such as potato, wheat and maize [18,19]. Therefore, the main objective of the present study was to evaluate the microbiological biosafety and agronomic significance of bacterial species associated with raw vegetables. For this purpose, enriched and selective culture media were used to isolate general and specific bacterial species associated with fresh produce. The final taxonomic status of the bacterial isolates was confirmed by 16S rRNA gene sequencing [20]. Fresh produce-associated microbial hazards have not been uncovered in detail in Pakistan. Therefore, in this study, bacterial diversity associated with carrot, cabbage and turnip was targeted. The main reason to work with these agronomic plants was their poor handling from field to local markets. This may have resulted in their cross contamination with potential human pathogens during transportation. It was also hypothesized that after interacting with host plants, bacterial species may have developed beneficial plant growth-promoting traits. With this in mind, the phylogenetic diversity and agricultural significance of bacterial communities were also inspected to identify the range of bacterial species showing positive interactions with colonized plants. 

## 2. Materials and Methods

### 2.1. Sample Collection and Culture Media

Samples of carrot, cabbage and turnip (12 samples each) were collected from different sites in Lahore, Pakistan. Samples were collected from 12 randomly selected sites that were comprised of small vegetable shops, major vegetable markets and superstores. Samples were collected in sterile plastic bags and brought to the laboratory and processed within 24 h. In order to assess the bacterial diversity, different types of culture media including Luria–Bertani Agar (L-agar), Baird–Parker Agar (BPA), Eosin Methylene Blue Agar (EMB), Mannitol Salt Agar (MSA), MacConkey Agar (MAC) and Xylose Lysine Deoxycholate Agar (XLD) were used. L-agar is extensively used as a general and non-selective growth medium [21]. BPA and MSA are selective for the growth of Gram-positive bacteria specifically for *Staphylococcus aureus*. EMB, MAC and XLD support the growth of Gram-negative bacteria. EMB agar, in particular, differentiates between lactose and non-lactose fermenter groups of Gram-negative bacteria and is widely used for the isolation of *E. coli*, *Citrobacter freunddii* and *Enterobacter cloacae*. XLD agar is particularly used for the isolation of *Salmonella enterica* and *Shigella flexneri* from food and clinical samples [22]. 

### 2.2. Isolation of Bacterial Strains

Samples were crushed in sterile mortars and pestles and diluted serially before spreading. For serial dilution, 10 g of the crushed sample was added into pre-autoclaved 100 mL normal saline solution (0.85% NaCl) and mixed well. Approximately 50 µL of this dilution was aseptically plated on the surface of L-agar and other selective media as mentioned above. After spreading, the plates were incubated between 30 to 37 °C for 24 h. Following incubation, colonies with different morphologies were selected and purified using several rounds of streaking on respective media (Appendix A). The purity of the colonies was also monitored by performing Gram staining.

### 2.3. 16S rRNA Gene Sequencing

For 16S rRNA gene sequencing, genomic DNA was extracted from bacterial cultures grown overnight using a Genomic DNA Purification Kit (Promega, USA). An approximately 1·5-kb DNA fragment containing 16S rRNA gene was amplified using forward 27f and reverse primer 1522r [23]. PCR amplification was performed using 50 µL of Dream Taq^TM^ Green PCR Master Mix (ThermoFisher Scientific, Chelmsford, MA, USA) with 0.5 µg of chromosomal DNA template and 0.5 µM of each primer. The reaction mixtures were incubated in a thermocycler Primus 96 (PeQLab, Erlangen, Germany) at 94 °C for 5 min and passed through 30 cycles: denaturation for 20 s at 94 °C, primer annealing for 20 s at 50 °C and extension at 72 °C for 2 min. Final extension was carried out at 72 °C for 5 min. The amplified products were purified using a QIAquick Gel Extraction Kit (QIAGEN, Frederick, MD, USA) and the samples were sent to First BASE laboratories (Singapore) for the sequencing of 16S rRNA gene. 

### 2.4. Phylogenetic Analysis

The sequences were analyzed using the bioinformatics tool, CHROMAS lite version 2.4.1.0, and homology was searched for in the NCBI Basic Local Alignment Search Tool (BLAST) for the identification of bacterial strains. After identification, all the sequences were aligned with a multiple sequence alignment program (ClustalW) using MEGA 6.05 software [24]. The phylogenetic relationships among different bacterial genera were studied after constructing phylogenetic trees using the neighbor-joining (NJ) method [25]. 

### 2.5. Antibiotic Susceptibility Pattern of Bacterial Strains

The antibiotic sensitivity pattern of purified bacterial strains was evaluated by following the method of Bauer et al. [26]. Plates of Mueller–Hinton agar were prepared and inoculated with bacterial strains using a sterile cotton swab to ensure confluent growth. The antibiotic susceptibility test discs (Bioanalyse^®^, Ankara, Turkey) for gentamicin (30 µg), tobramycin (10 µg), amikacin (30 µg), cephalexin (30 µg), chloramphenicol (30 µg), amoxicillin (25 µg), nalidixic acid (30 µg) and tetracycline (30 µg) were aseptically placed to the surface of agar plates at well-spaced intervals. Three sets of plates for each strain or antibiotic were prepared for analysis. The plates were then incubated at 37 °C for 24 h. After incubation, the plates were observed for the presence of clear zones of inhibition around the antibiotic disc. Zones were measured in millimeters (mm) using the Inhibition Zone Ruler provided by the manufacturer. The zones were then compared with the standardized chart for antibiotics (M100-S23) given by the clinical laboratory standard institute (CLSI, 2013). 

### 2.6. Functional Diversity of Plant-Associated Bacteria

Production of auxin in 25 mL of Luria–Bertani broth (L-broth) medium was detected in the presence of 0 or 500 µg mL^−1^ of L-tryptophan. Flasks were inoculated with purified bacterial cultures in triplicate (adjusted to 10^7^ CFUs per mL) and incubated on a shaker in the dark at 37 °C for 72 h. After incubation, the cultures were centrifuged (Sigma 1-14, Osterode, Germany) at 18,900 relative centrifugal field (RCF) for 10 min. One milliliter of supernatant was taken in a test tube and 2 mL of Salkowski’s reagent was added [27]. The tubes were kept in the dark for 30 min for the development of pink to red coloration. Approximately 300 µL of each sample was taken from the test tube and added into the wells of a microtiter plate. The intensity of the color was measured at 535 nm using a microplate spectrophotometer (Epoch, BioTek, Winooski, VT, USA). Finally, the concentrations of auxin produced by different isolates were determined by comparing the values with the standard curve. The standard curve was constructed using different concentrations of standard auxin (5 to 200 µg mL^−1^) in 1 mL of distilled water and processed for colorimetric analysis as mentioned above. Phosphate solubilization ability of the bacterial isolates was determined by streaking strains on Pikovskaya agar medium [28]. Hydrogen cyanide (HCN) activity of bacterial strains was determined as mentioned by Ahmad et al. [29]. 

### 2.7. Biofilm Formation

To determine the biofilm-forming ability of purified bacterial isolates the method by Christensen et al. [30] was followed with slight modifications. Bacterial strains were streaked on Tryptic Soy Agar (TSA) plates and incubated at 37 °C for 24 h. From the TSA plates, the cultures were picked and inoculated in Tryptic Soy Broth (TSB) medium and incubated as mentioned above. Following incubation, the broth cultures were standardized (optical density adjusted to 0.2 at 600 nm) and 20 µL of each culture was transferred to the wells of a 96-well flat bottom microtiter plate (Orange Scientific) that already contained 180 µL of TSB medium. Negative controls with 200 µL of TSB medium were also kept for comparison. The assay was performed in triplicate for all treatments. To promote biofilm formation, the plates were incubated aerobically on a shaker at 37 °C for 72 h. After incubation, the well contents were discarded and washed thrice with 250 µL of sterile distilled water to remove any non-adherent and weakly adherent cells. Plates were air dried for 30 min. The biofilm formed in the wells was fixed with 250 µL/well of 98% methanol for 15 min. After air drying, the fixed bacterial cells were stained with 200 µL of 0.1% v/v crystal violet solution for 5 min. The excessive stain was removed by placing the plate under slow running tap water and the plate was air dried. Re-solubilization of crystal violet with the adherent cells was done by adding 200 µL/well of 33% v/v glacial acetic acid. The optical density (OD) was measured at 570 nm using a microplate spectrophotometer (Epoch, BioTek, Winooski, VT, USA). 

### 2.8. Statistical Analysis

Data for bacterial auxin production and biofilm formation was subjected to an analysis of variance (ANOVA) using SPSS 16 program. The means of different treatments were separated using Duncan’s multiple range test (*p* ≤ 0.05). 

## 3. Results

### 3.1. 16S rRNA Gene Sequencing

The sequences obtained from First BASE laboratories were analyzed and refined by trimming the noise from both ends using the bioinformatics tool, CHROMAS lite version 2.4.1.0 (Sequences Appendix A included). These sequences were used to search for homology with already identified sequences in NCBI using the default nucleotide Basic Local Alignment Search Tool (BLAST) settings. After comparison, the majority of the strains showed up to 99% similarity with their respective identified species. Analysis showed that the strains belong to Bacillus, Staphylococcus, Exiguobacterium, Lysinibacillus, Arthrobacter, Burkholderia, Acinetobacter, Pseudomonas, Stenotrophomonas, Serratia, Klebsiella, Citrobacter, Enterobacter, Pantoea and Kluyvera genera. After identification, the sequences were submitted in Genbank under accession numbers KJ865549 to KJ865603 (Table 1, Table 2 and Table 3). 

### 3.2. Phylogenetic Analysis

To study the phylogenetic relationships among bacterial strains, the sequences were first aligned with the multiple sequence alignment program (ClustalW) using MEGA 6 software. The phylogenetic tree was constructed using the neighbor-joining method to show evolutionary relationships among different isolates (Figure 1). In the phylogenetic tree, the first cluster was comprised of different strains of *Bacillus*. The second cluster includes *Staphylococcus equorum* BPt-5, *Staph. gallinarum* MSt-3, *Staph. warneri* MSc-5, *Staph. arlettae* MCb-3, *Staph. xylosus* Lt-41, *Staph. aureus* BPb-5, *Staph. xylosus* Xt-1, *Staph. warneri* Lt-73 and *Staph. xylosus* MSt-1. Similarly, different strains of *Acinetobcter* (*Ac. bouvetii* EMt-1, *Ac. calcoaceticus* MSt-6, *Ac. calcoaceticus* Mc-4, *Ac. calcoaceticus* Lc-52, *Ac. colcoaceticus* BPb-3, *Ac. colcoaceticus* Eb-4, *Ac. colcoaceticus* EMc-4, *Ac. colcoaceticus* Eb-8) covered another large cluster in the tree. Additionally, different strains of Gram-negative genera occupied the lower portion of the phylogenetic tree. 

### 3.3. Bacterial Diversity of Fresh Vegetables

The microbiological analysis of carrot showed the association of 18 bacterial strains that belong to 11 bacterial genera (Table 1). The highest number for colonization was observed for the genera *Serratia*, *Bacillus* and *Acinetobacter*. For cabbage, 19 bacterial strains that represented nine bacterial genera were detected (Table 2). Maximum colonization was shown by the genus *Bacillus* followed by *Acinetobacter* and *Staphylococcus*. Additionally, some strains that specifically colonized cabbage include *Ar. nicotianae* Lb-41, *Ex. mexicanum* MCb-4, *K. pneumoniae* EB-1 and *Bur. cepacia* Eb-6. In the case of turnip, maximum colonization was shown by the genus *Staphyloccous* followed by *Bacillus* and *Enterobacter* (Table 3). Among the 18 strains that were detected from turnip, *Kluyvera cryocrescens* MCt-5 was specifically associated with this vegetable. Overall, the strains from the genus *Bacillus*, *Staphylococcus*, *Serratia* and *Acinetobacter* were frequently associated with the surfaces of carrot, turnip and cabbage, whereas, *Lysinibacillus, Stenotrophomonas*, *Citrobacter* and *Enterobacter* were commonly associated with carrot or turnip. Some bacterial genera were found to specifically colonize cabbage (*Exiguobacterium*, *Arthrobacter*, *Burkholderia* and *Klebsiella*), turnip (*Kluyvera*) or carrot (*Pseudomonas* and *Pantoea*). 

### 3.4. Antibiotic Susceptibility of Bacterial Strains

The susceptibility of all the identified plant-associated bacterial isolates towards different antibiotics was determined. Figure 2 shows the sensitivity pattern of *B. cereus* LCw-22 and *A. calcoaceticus* MSt-6. Out of 55 strains, 30 and 27 strains were resistant against amoxicillin and nalidixic acid, respectively. Both of these drugs are broad-spectrum antibiotics which are equally effective for Gram-positive and Gram-negative groups of bacteria (Table 4 and Table 5). Resistance against tobramycin, gentamicin and amikacin were shown by 18, eight and two bacterial strains, respectively. On the other hand, 24 bacterial strains were resistant to cephalexin. Chloramphenicol and tetracycline are also broad-spectrum antibiotics and resistance against them was depicted, respectively, by seven and five strains while others recorded sensitive or intermediate results. For Gram-positive bacteria, the majority of the strains recorded sensitivity against amikacin, gentamicin, tetracycline and chloramphenicol (Table 4). *St. maltophilia* Mc-3 showed the highest resistance against the applied antibiotics as out of eight, it recorded resistant against seven antibiotics (Table 5). Similarly, *B. subtilis* MCb-8, *Ent. cloacae* PCt-2, *Ent. amnigenus* EMt-5 and *Staph. warneri* Lt-73 showed resistance for six, five, five and five antibiotics, respectively. 

### 3.5. Functional Diversity of Bacteria

Bacterial isolates were evaluated for their auxin production ability in the presence of 500 µg mL^−1^ L-tryptophan. The highest levels of auxin were detected for *B. cereus* PCt-1, *B. cereus* MSb-3 and *Se. ureilytica* MCt-6, which recorded 5-, 4- and 4-fold increases, respectively, compared to the un-amended control. A significant 3-fold increase was observed with each of the strains: *Pan. vagans* Eb-2, *Se. rubidaea* EMc-2, *Ac. calcoaceticus* Eb-4, *Se. rubidaea* Mc-2 and *Cit. freundii* Xc-6. The maximum auxin concentrations of 36, 34 and 33 µg mL^−1^ were obtained by *L. fusiformis* Xt-6, *Ent. cloacae* PCt-2 and *K. pneumoniae* Eb-1, respectively, in the absence of L-tryptophan. However, in the L-tryptophan-amended medium, *St. maltophilia* MCt-1 (101 µg mL^−1^), *B. cereus* PCt-1 (97 µg mL^−1^), *Se. rubidaea* Mc-2 (77 µg mL^−1^), *K. pneumoniae* Eb-1 (75 µg mL^−1^) and *Ent. cloacae* PCt-2 (71 µg mL^−1^) were the most promising for in vitro auxin production (Figure 3). 

For HCN production, *Pan. dispersa* BPc-3 and *B. subtilis* Lb-61 were strongly positive and turned the color of the filter paper to a dark orange (Appendix A). Similarly, *Ac. calcoaceticus* Eb-4, *Ac. calcoaceticus* Eb-8, *Ac. calcoaceticus* BPb-3, *K. pneumoniae* Eb-1, *B. subtilis* MCb-8 and *B. cereus* MSb-3 produced clear zones after inoculation on Pikovskaya media, which indicated a positive test for phosphate solubilization (Appendix A). 

### 3.6. Biofilm Formation

The biofilm-forming potential of purified bacterial isolates was examined by a microtiter plate assay. After 72 h of incubation at 37 °C in a 96-well microtiter plate, the cells were stained with crystal violet and the cell biomass was recorded by a spectrophotometer. The strains giving a cell mass OD equal to or above 1 were considered good biofilm producers (Figure 4). The bacterial strains *Ar. nicotianae* Lb-41, *Staph. arlettae* MCb-3, *Ex. mexicanum* MCb-4 and *Staph. xylosus* Xt-1 were observed as strong biofilm producers. Similarly, good biofilm production was also noted with *K. pneumoniae* Eb-1, *Ac. calcoaceticus* Eb-8, *Se. rubidaea* LCr-22, *Staph. xylosus* Lt-41, *B. cereus* LCw-22, *Kl. ryocrescens* MCt-5, *Se. rubidaea* Mc-2, *B. anthracis* MSb-4 and *Cit. werkmanii* Xt-3. The bacterial strains also showed variations in their ability to form an in vitro biofilm. For instance, *K. pneumoniae* Eb-1, *Ac. calcoaceticus* Eb-8, *B. anthracis* MSb-4 and *A. nicotianae* Lb-41 that were associated with cabbage recorded good potential for biofilm formation as compared to other crop isolates. For carrot, three bacterial strains that include *B. cereus* LCw-22 and *Se. rubidaea* (Mc-2, Lcr-22) showed significant biofilm formation. In the case of turnip, *Staph. xylosus* (Lt-41, Xt-1) was found to be very effective as a biofilm producer. 

## 4. Discussion

Fresh vegetables are colonized by thousands of microbial species which may or may not be a part of their natural microbiome. Extraneous microorganisms can be found associated with plants due to contamination from different sources such as unclean irrigation water, organic fertilizers, animal and human wastes [10]. If pathogenic, these microorganisms can influence human health in several ways and thus present serious food safety challenges. A number of studies have already explored fresh vegetables for specific plant-associated human pathogens [31,32,33]. The present study demonstrated the biosafety concerns associated with raw-eaten fresh vegetables from some areas of Lahore, Pakistan. Moreover, little is known about the microbiological hazards associated with fresh agricultural produce in Pakistan. Therefore, in the present work, we focused on the sanitation of local vegetable markets with concurrent screening for beneficial bacteria–plant interactions. 

Although no obligate human pathogens were detected in our vegetable samples, some potentially pathogenic bacteria that were isolated include *B. cereus*, *B. anthracis*, *Staph. aurues*, *Ent. cloacae*, *Ent. amnigenus* and *K. Pneumoniae*. The presence of such organisms in fresh raw-eaten vegetables is a serious concern for the consumer’s health. *B. cereus* is a common foodborne human pathogen that causes food poisoning by producing toxins in food. It has been reported to be present in the intestinal tract of mammals and their waste material [34]. *B. cereus* can cause two distinct types of food poisoning that includes diarrheal or emetic syndrome. Nonhemolytic enterotoxin has been shown to be associated with diarrheal syndrome. It is also responsible for a variety of local and systemic infections [35]. *Ent. cloacae, Ent. amnigenus* and *K. Pneumoniae* are members of the Enterobacteriaceae family and their incidence in vegetables is also linked to the fecal material of warm-blooded animals. *Enterobacter* is also a member of the coliform group of bacteria which share their growth properties with human pathogenic bacteria. Their presence clearly indicates the possible presence of pathogens in these vegetables and their consumption without proper precautions can impact human health badly. A study by Falomir et al. [36] also reported the isolation of *Ent. cloacae* and *K. pneumoniae* from fresh vegetables. Staphylococcal enterotoxins (SEs) are a major cause of food poisoning, which typically occurs after ingestion of different contaminated food products. It has been shown that SEA and SEH enterotoxins are the most common cause of staphylococcal food poisoning around the world [37]. *B. cereus*, *E. coli*, *Listeria monocytogenes* and *Salmonella* are considered to be the most frequent bacterial pathogens associated with fresh produce-related outbreaks [38]. In most of the cases, contaminated vegetables with the members of Enterobacteriaceae were associated with gastrointestinal diseases such as diarrhea [39]. Similarly, produce-related pathogenic organisms such as *Clostridium botulinum*, *B. cereus* and *S. aureus* produce heavy amounts of toxins during their colonization and are the major causes of food poisoning [40]. 

As the isolation of some potential pathogens from fresh vegetables made the safety of these products questionable, it was necessary to screen all the identified microorganisms for multidrug resistance. High resistances of 52 % (amoxicillin) and 59 % (nalidixic acid) were observed with two broad-spectrum antibiotics used for Gram-positive bacteria (Table 4). Antibiotics used against Gram-negative bacteria were effective for the majority of the strains (Table 5). However, 57% and 50% resistance was recorded for amoxicillin and cephalexin, respectively. In the present study, a few strains of *B. cereus* (LCw-22, MSt-7, MSb-3) and *Pantoea* sp. MSc-1 were sensitive to the tested drugs. This may show the potential of these drugs in bacterial disease prevention. Nevertheless, multiple resistance against amoxicillin, tetracycline, gentamicin, chloamphenicol and other chemotherapeutic agents have been reported in *Enterobacter* and *Klebsiella* species isolated from fresh vegetables [36]. 

In this study, bacterial strains associated with fresh agricultural produce also exhibited beneficial plant growth-promoting attributes; especially, auxin production and biofilm formation. A variety of IAA-producing bacterial strains has been shown to harbor by plants that positively influence plant growth and productivity [41,42,43,44]. In the present study, supplementation of L-broth with L-tryptophan enhanced auxin production several folds compared to the control. For instance, *Se. rubidaea* EMc-2 recorded the lowest production of auxin (12.24 µg mL^−1^) in the absence of L-tryptophan. However, supplementation of the medium with L-tryptophan resulted in a 2-fold increase in auxin content in liquid culture supernatants (Appendix A). L-tryptophan has been considered as precursor for auxin biosynthesis in plants as well as for microbes [20]. The results of present study may indicate the ability of microbes to use the natural source of L-tryptophan from root exudates or soil to produce phytohormones within the plant’s rhizosphere. In this way, microbes may provide an exogenous source of phytohormones to plants, especially during the early stages of development [45]. A variety of bacterial genera can interact and colonize plant surfaces by the formation of biofilms. Biofilm formation on plant surfaces may be associated with symbiotic or pathogenic response depending on the microbial species [46]. Our results showed the colonization of fresh agricultural produce by a few potential human pathogens that belong to the genera of *Bacillus*, *Enterobacter*, *Staphylococcus* and *Klebsiella*. In present study, *K. pneumonia* Eb-1 and *B. anthracis* MSb-1 that were associated with cabbage showed significant biofilm formation (Figure 4). Thus, these opportunistic human pathogens can pose severe health hazards for consumers. Agricultural produce may be contaminated by animal droppings, farmworkers’ hands and irrigation water, etc. Fresh produce-related outbreaks can be controlled my managing agricultural and post-harvest practices [47]. The colonization of agronomically important plants by opportunistic human pathogens has been reported. For instance, the presence of *B. cereus*, *P. aeruginosa* and *S. saprophyticus* has been reported within the root system of plants with several plant growth-promoting attributes including IAA [48,49,50]. It has been reported that pathogenic species to animals and humans are mainly transmitted through the food chain. Therefore, pathogenic bacteria can contaminate plant surfaces and actively interact and colonize them as an alternate host [10]. Moreover, in previous studies, these bacteria have also been shown to exhibit beneficial plant attributes including IAA [51]. Bacterial genera such as *Burkholderia*, *Enterobacter*, *Stenotrophomonas*, *Pseudomonas*, *Bacillus* and *Staphylococcus* are associated with plants and affect their hosts in beneficial ways [11]. They have also been reported to stimulate plant growth by the production of IAA, biofilm formation or phosphate solubilization [20,46]. Thus, bacterial strains showing beneficial traits may be used as biofertilizers to minimize the use or cost of inorganic fertilizers for crop production. Phosphorus (P) is the second most important macronutrient (after nitrogen) required for plant growth and development. Inadequate P availability may cause stunted plant growth, and an abnormal dark green leaf color with reddish to purple tips or margins. Rhizobacteria can secrete organic acids or phosphatases in soils to convert insoluble P into soluble ions that can be utilized by plants [52]. In the current study, *Pan. dispersa* BPc-3 and *B. subtilis* Lb-61 were strongly positive for HCN (Appendix A). Production of HCN by bacteria may play a very critical role in the suppression of phytopathogens [49]. 

## 5. Conclusions

In conclusion, fresh raw vegetables were colonized by a variety of bacterial genera. Although no obligate human pathogens were detected, the presence of a few members of potential human pathogens makes the biosafety of these vegetable questionable. They can cause harmful health effects to consumers. It may be difficult to completely remove these bacteria from food but certain precautionary measures particularly in agricultural practices, harvesting and processing can decrease the number of potential risk factors in these fresh produce products. Nevertheless, due to the close proximity with the plant surfaces, these microbes also harbor beneficial plant growth-promoting traits such as auxin production, mineral solubilization and biofilm formation. Overall, this study reported the association of several bacterial genera from the surfaces of agronomically important raw vegetables. The methodological strategies used in this study will help to investigate or screen general or specific bacterial diversity associated with other raw vegetables. In future, this study can be further extended to identify the indigenous bacterial communities associated with fresh agricultural produce growing in different geographical locations.

## Figures and Tables

**Figure 1 plants-08-00091-f001:**
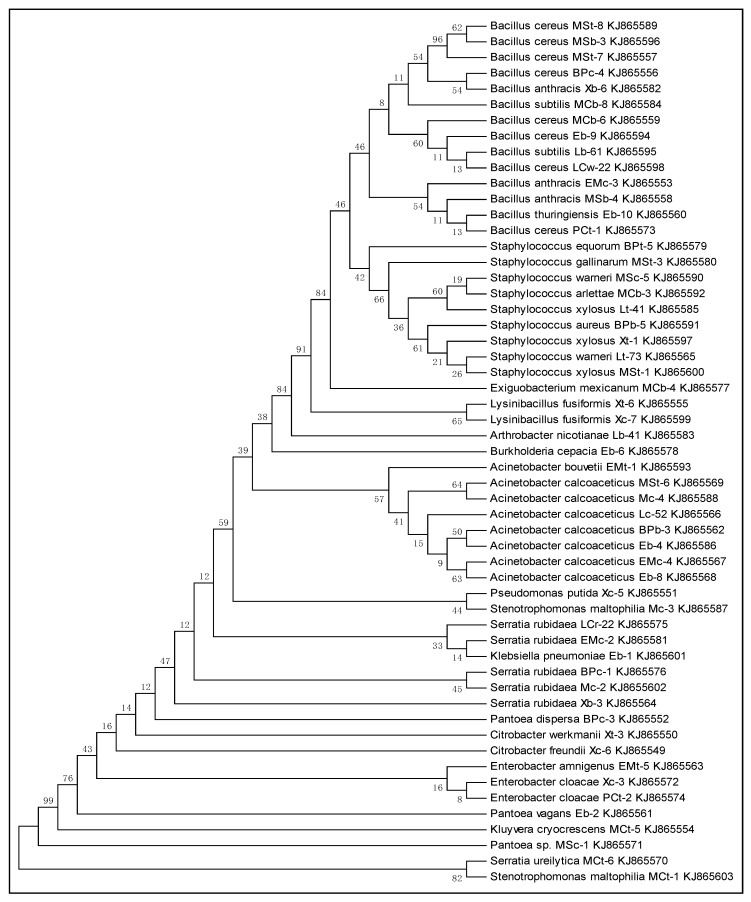
Phylogenetic analysis of 55 bacterial isolates associated with the surfaces of fresh vegetables (carrot, cabbage and turnip). Nucleotide sequences were trimmed after 16S rRNA gene sequencing. Phylogenies were inferred using the neighbor-joining method and the tree was constructed using MEGA 6 [24]. Numbers at the branch points indicate the percentage of 1000 bootstrap resampling.

**Figure 2 plants-08-00091-f002:**
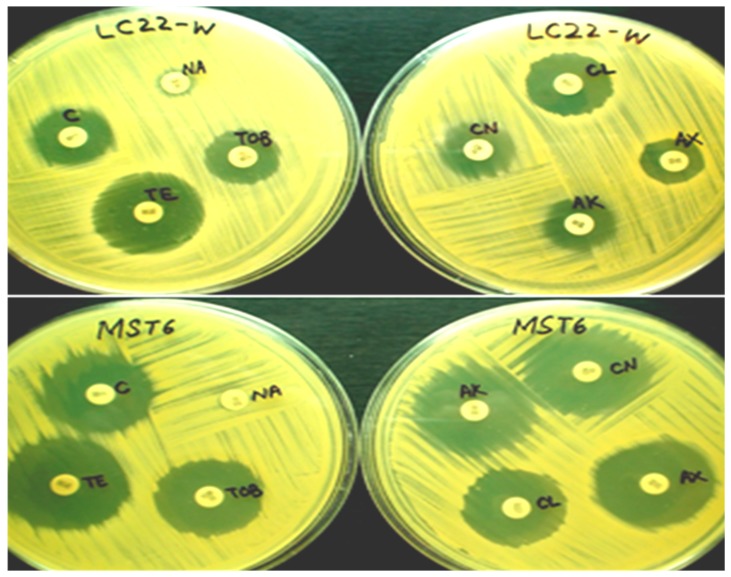
Antibiotic susceptibility pattern of purified strains of *B. cereus* LCw-22 and *A. calcoaceticus* MSt-6. Plates were incubated at 37 °C for 24 h. Abbreviations: C = chloramphenicol, NA = nalidixic acid, TE = tetracycline, TOB = tobramycin, AK = amikacin, CN = gentamicin, CL = cephalexin, AX = amoxicillin.

**Figure 3 plants-08-00091-f003:**
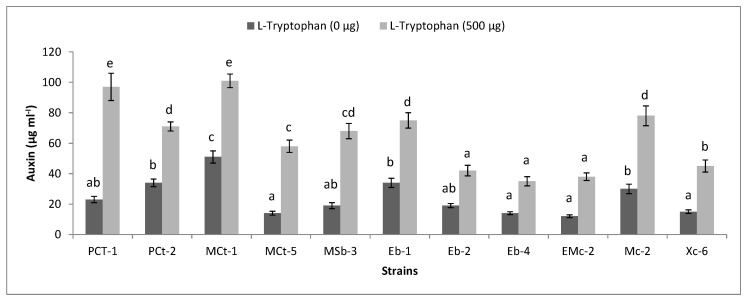
Auxin production by some representative purified bacterial strains in the presence and absence of L-tryptophan. Strains were grown in Luria–Bertani broth (L-broth) at 37 °C for 72 h. Bar represents the mean ± S.E. of three replicates. Different letters on bars indicate significant differences between respective treatments using Duncan’s multiple range test (*p* ≤ 0.05).

**Figure 4 plants-08-00091-f004:**
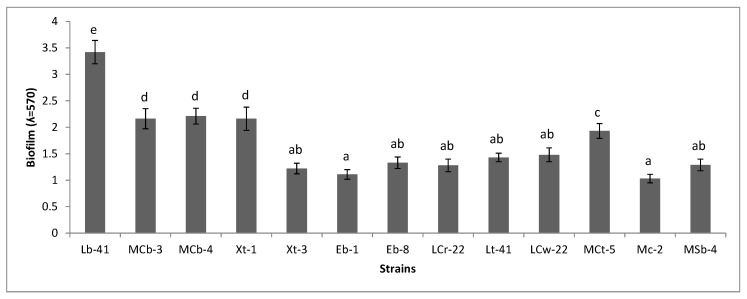
Biofilm formation by some selected purified bacterial strains. Biofilm assay was performed in Tryptic Soy Broth (TSB) medium at 37 °C. Bar represents the mean ± S.E. of three replicates. Different letters on bars indicates significant differences between treatments using Duncan’s multiple range test (*p* ≤ 0.05).

**Table 1 plants-08-00091-t001:** 16S rRNA gene sequencing of bacteria isolated from carrot at different temperatures and in different culture media.

Serial No.	Isolates	Temperature for Isolation	Culture Media	Identified as	Accessions
1	BPc-4	30 °C	L-agar	*Bacillus cereus* BPc-4	KJ865556
2	EMc-3	37 °C	L-agar	*B. anthracis* EMc-3	KJ865553
3	LCw-22	30 °C	L-agar	*B. cereus* LCw-22	KJ865598
4	MSc-5	37 °C	MSA	*Staphylococcus warneri* MSc-5	KJ865590
5	Xc-7	30 °C	L-agar	*Lysinibacillus fusiformis* Xc-7	KJ865599
6	BPc-1	37 °C	MAC	*Serratia rubidaea* BPc-1	KJ865576
7	BPc-3	37 °C	MAC	*Pantoea dispersa* BPc-3	KJ865552
8	EMc-2	37 °C	MAC	*Se. rubidaea* EMc-2	KJ865581
9	EMc-4	37 °C	L-agar	*Acinetobacter calcoaceticus* EMc-4	KJ865567
10	Lc-52	37 °C	L-agar	*Ac. calcoaceticus* Lc-52	KJ865566
11	Lcr-22	37 °C	MAC	*Se. rubidaea* Lcr-22	KJ865575
12	Mc-2	37 °C	MAC	*Se. rubidaea* Mc-2	KJ865602
13	Mc-3	37 °C	L-agar	*Stenotrophomonas maltophilia* Mc-3	KJ865587
14	Mc-4	37 °C	L-agar	*Ac. calcoaceticus* Mc-4	KJ865588
15	MSc-1	37 °C	MAC	*Pantoea sp.* MSc-1	KJ865571
16	Xc-3	37 °C	EMB	*Enterobacter cloacae* Xc-3	KJ865572
17	Xc-5	30 °C	L-agar	*Pseudomonas putida* Xc-5	KJ865551
18	Xc-6	37 °C	EMB	*Citrobacter freundii* Xc-6	KJ865549

Abbreviations: L-agar = Luria–Bertani Agar, MSA = Mannitol Salt Agar, MAC = MacConkey Agar, EMB = Eosine Methylene Blue agar.

**Table 2 plants-08-00091-t002:** 16S rRNA gene sequencing of bacteria isolated from cabbage at different temperatures and in different culture media.

Serial No.	Isolates	Temperature for Isolation	Culture Media	Identified as	Accessions
1	BPb-5	37 °C	MSA	*Staph. aureus* BPb-5	KJ865591
2	Eb-9	30 °C	L-agar	*B. cereus* Eb-9	KJ865594
3	Eb-10	30 °C	L-agar	*B. thuringiensis* Eb-10	KJ865560
4	Lb-41	30 °C	L-agar	*Arthrobacter nicotianae* Lb-41	KJ865583
5	Lb-61	30 °C	L-agar	*B. subtilis* Lb-61	KJ865595
6	MCb-3	37 °C	L-agar	*Staph. arlettae* MCb-3	KJ865592
7	MCb-4	37 °C	L-agar	*Exiguobacterium mexicanum* MCb-4	KJ865577
8	MCb-6	30 °C	L-agar	*B. cereus* MCb-6	KJ865559
9	MCb-8	30 °C	L-agar	*B. subtilis* MCb-8	KJ865584
10	MSb-3	30 °C	L-agar	*B. cereus* MSb-3	KJ865596
11	MSb-4	37 °C	L-agar	*B. anthracis* MSb-4	KJ865558
12	Xb-6	30 °C	L-agar	*B. anthracis* Xb-6	KJ865582
13	BPb-3	37 °C	L-agar	*Ac. calcoaceticus* BPb-3	KJ865562
14	Eb-1	37 °C	MAC	*Klebsiella pneumoniae* Eb-1	KJ865601
15	Eb-2	37 °C	MAC	*Pa. vagans* Eb-2	KJ865561
16	Eb-4	30 °C	L-agar	*Ac. calcoaceticus* Eb-4	KJ865586
17	Eb-6	30 °C	L-agar	*Burkholderia cepacia* Eb-6	KJ865578
18	Eb-8	37 °C	L-agar	*Ac. calcoaceticus* Eb-8	KJ865568
19	Xb-3	37 °C	MAC	*Se. rubidaea* Xb-3	KJ865564

**Abbreviations**: L-agar = Luria–Bertani Agar, MSA = Mannitol Salt Agar, MAC = MacConkey Agar, EMB = Eosine Methylene Blue agar.

**Table 3 plants-08-00091-t003:** 16S rRNA gene sequencing of bacteria isolated from turnip at different temperatures and in different culture media.

Serial No.	Isolates	Temperature for Isolation	Culture Media	Identified as	Accessions
1	BPt-5	37 °C	MSA	*Staph. equorum* BPt-5	KJ865579
2	Lt-41	37 °C	MSA	*Staph. xylosus* Lt-41	KJ865585
3	Lt-73	37 °C	MSA	*Staph. warneri* Lt-73	KJ865565
4	MSt-1	37 °C	MSA	*Staph. xylosus* MSt-1	KJ865600
5	MSt-3	37 °C	MSA	*Staph. gallinarum* MSt-3	KJ865580
6	MSt-7	30 °C	L-agar	*B. cereus* MSt-7	KJ865557
7	MSt-8	30 °C	L-agar	*B. cereus* MSt-8	KJ865589
8	PCt-1	30 °C	L-agar	*B. cereus* PCt-1	KJ865573
9	Xt-1	37 °C	MSA	*Staph. xylosus* Xt-1	KJ865597
10	Xt-6	30 °C	L-agar	*L. fusiformis* Xt-6	KJ865555
11	EMt-1	37 °C	L-agar	*Ac. bouvetii* EMt-1	KJ865593
12	EMt-5	37 °C	EMB	*E. amnigenus* EMt-5	KJ865563
13	MCt-1	37 °C	L-agar	*St. maltophilia* MCt-1	KJ865603
14	MCt-5	37 °C	MAC	*Kluyvera cryocrescens* MCt-5	KJ865554
15	MCt-6	37 °C	MAC	*Se. ureilytica* MCt-6	KJ865570
16	MSt-6	37 °C	L-agar	*Ac. calcoaceticus* MSt-6	KJ865569
17	PCt-2	37 °C	EMB	*E. cloacae* PCt-2	KJ865574
18	Xt-3	37 °C	EMB	*C. werkmannii* Xt-3	KJ865550

**Abbreviations**: L-agar = Luria–Bertani Agar, MSA = Mannitol Salt Agar, MAC = MacConkey Agar, EMB = Eosine Methylene Blue agar.

**Table 4 plants-08-00091-t004:** Antibiotic susceptibility pattern of different Gram-positive bacterial isolates. For analysis, three set of plates for each strain or antibiotic were incubated at 37 °C for 24 h.

Strains	Antibiotics							
	AK	AX	CL	CN	NA	TOB	TE	C
	Zones of Inhibition (mm) *							
*Bacillus cereus* BPc-4	18 (S)	10 (R)	8 (R)	14 (I)	14 (I)	15 (S)	22 (S)	10 (R)
*Staphylococcus equorum* BPt-5	26 (S)	24 (R)	11 (R)	20 (S)	0 (R)	16 (S)	16 (I)	26 (S)
*S. aureus* BPb-5	18 (S)	8 (R)	18 (S)	18 (S)	14 (I)	16 (S)	26 (S)	24 (S)
*B. anthracis* EMc-3	20 (S)	16 (I)	14 (I)	24 (S)	16 (I)	24 (S)	24 (S)	24 (S)
*B. cereus* Eb-9	16 (S)	20 (S)	0 (R)	22 (S)	20 (S)	12 (R)	18 (I)	24 (S)
*B. thuringiensis* Eb-10	18 (S)	20 (S)	0 (R)	14 (I)	0 (R)	12 (R)	20 (S)	12 (R)
*S. xylosus* Lt-41	20 (S)	24 (R)	22 (S)	20 (S)	0 (R)	18 (S)	22 (S)	20 (S)
*S. warneri* Lt-73	20 (S)	0 (R)	0 (R)	12 (R)	0 (R)	10 (R)	16 (I)	24 (S)
*Arthrobacter nicotianae* Lb-41	22 (S)	26 (S)	16 (S)	16 (S)	0 (R)	14 (I)	20 (S)	30 (S)
*B. subtilis* Lb-61	28 (S)	20 (S)	14 (I)	30 (S)	12 (R)	22 (S)	26 (S)	14 (I)
*B. cereus* LCw-22	16 (I)	14 (I)	20 (S)	16 (S)	10 (R)	16 (S)	24 (S)	18 (S)
*S. arlettae* MCb-3	32 (S)	14 (I)	22 (S)	28 (S)	0 (R)	22 (S)	8 (R)	24 (S)
*Exiguobacterium mexicanum* MCb-4	22 (S)	38 (S)	28 (S)	20 (S)	18 (I)	16 (S)	26 (S)	24 (S)
*B. cereus* MCb-6	20 (S)	14 (I)	18 (S)	16 (S)	0 (R)	14 (I)	24 (S)	16 (I)
*B. subtilis* MCb-8	16 (I)	0 (R)	0 (R)	10 (R)	12 (R)	10 (R)	16 (I)	10 (R)
*S. xylosus* MSt-1	24 (S)	22 (R)	10 (R)	20 (S)	0 (R)	20 (S)	20 (S)	24 (S)
*S. gallinarum* MSt-3	14 (I)	12 (R)	16 (S)	14 (I)	0 (R)	10 (R)	20 (S)	18 (S)
*B. cereus* MSt-7	26 (S)	14 (I)	32 (S)	24 (S)	18 (I)	18 (S)	14 (R)	20 (S)
*B. cereus* MSt-8	18 (S)	8 (R)	14 (I)	12 (R)	16 (I)	18 (S)	24 (S)	28 (S)
*B. cereus* MSb-3	34 (S)	42 (S)	24 (S)	30 (S)	12 (R)	18 (S)	24 (S)	30 (S)
*B. anthracis* MSb-4	14 (I)	16 (I)	12 (I)	14 (I)	14 (I)	20 (S)	18 (I)	20 (S)
*S. warneri* MSc-5	24 (S)	40 (S)	38 (S)	30 (S)	14 (I)	24 (S)	28 (S)	24 (S)
*B. cereus* PCt-1	18 (S)	0 (R)	0 (R)	14 (I)	16 (I)	12 (R)	16 (I)	16 (I)
*S. xylosus* Xt-1	20 (S)	20 (R)	20 (S)	18 (S)	0 (R)	18 (S)	22 (S)	20 (S)
*Lysinibacillus fusiformis* Xt-6	16 (I)	0 (R)	0 (R)	14 (I)	14 (I)	12 (R)	18 (I)	20 (S)
*B. anthracis* Xb-6	22 (S)	12 (R)	14 (I)	16 (S)	12 (R)	16 (S)	20 (S)	20 (S)
*L. fusiformis* Xc-7	34 (S)	0 (R)	0 (R)	25 (S)	0 (R)	8 (R)	32 (S)	40 (S)

* Letters in parenthesis indicate the level of sensitivity of the respective antibiotics. Abbreviations: R = resistant, I = intermediate, S = sensitive. Antibiotics: AK = Amikacin; AX = Amoxicillin; CL = Cephalexin; CN = Gentamicin; NA = Nalidixic acid; TOB = Tobramycin; TE = Tetracycline; C = Chloramphenicol.

**Table 5 plants-08-00091-t005:** Antibiotic susceptibility pattern of different Gram-negative bacterial isolates. For analysis, three set of plates for each strain or antibiotic were incubated at 37 °C for 24 h.

Strains	Antibiotics							
	AK	AX	CL	CN	NA	TOB	TE	C
	Zone of Inhibition (mm) *							
*Serratia rubidaea* BPc-1	18 (S)	12 (R)	0 (R)	14 (I)	14 (I)	12 (R)	14 (R)	18 (S)
*Pantoea dispersa* BPc-3	22 (S)	14 (I)	14 (I)	18 (S)	12 (R)	14 (I)	20 (S)	12 (R)
*Acinetobacter calcoaceticus* BPb-3	16 (I)	14 (I)	12 (I)	18 (S)	14 (I)	20 (S)	26 (S)	22 (S)
*Klebsiella penumoniae* Eb-1	18 (S)	0 (R)	14 (I)	18 (S)	18 (I)	12 (R)	20 (S)	22 (S)
*P. vagans* Eb-2	20 (S)	22 (S)	20 (S)	20 (S)	18 (I)	22 (S)	22 (S)	10 (R)
*A. calcoaceticus* Eb-4	20 (S)	0 (R)	10 (R)	12 (R)	14 (I)	16 (S)	18 (I)	20 (S)
*Burkholderia cepacia* Eb-6	24 (S)	16 (I)	20 (S)	18 (S)	0 (R)	20 (S)	20 (S)	14 (I)
*A. calcoaceticus* Eb-8	18 (S)	8 (R)	0 (R)	16 (S)	16 (I)	14 (I)	20 (S)	12 (R)
*A. bouvetii* EMt-1	18 (S)	22 (S)	26 (S)	14 (I)	18 (I)	24 (S)	26 (S)	22 (S)
*Enterobacter amnigenus* EMt-5	18 (S)	0 (R)	0 (R)	12 (R)	12 (R)	10 (R)	16 (I)	22 (S)
*S. rubidaea* EMc-2	26 (S)	12 (R)	16 (S)	16 (S)	0 (R)	14 (I)	22 (S)	16 (I)
*A. calcoaceticus* EMc-4	28 (S)	14 (I)	22 (S)	16 (S)	16 (I)	18 (S)	18 (I)	20 (S)
*A. calcoaceticus* Lc-52	16 (I)	12 (R)	12 (I)	14 (I)	12 (R)	14 (I)	18 (I)	18 (S)
*S. rubidaea* Lcr-22	22 (S)	14 (I)	0 (R)	16 (S)	18 (I)	12 (R)	18 (I)	20 (S)
*Stenotrophomonas maltophilia* MCt-1	16 (I)	0 (R)	0 (R)	14 (I)	22 (S)	10 (R)	20 (S)	22 (S)
*Kluyvera cryocrescens* MCt-5	12 (R)	10 (R)	10 (R)	14 (I)	12 (R)	16 (S)	18 (I)	18 (S)
*S. ureilytica* MCt-6	14 (I)	10 (R)	14 (I)	12 (R)	14 (I)	16 (S)	22 (S)	20 (S)
*S. rubidaea* Mc-2	24 (S)	18 (S)	14 (I)	16 (S)	12 (R)	18 (S)	16 (I)	22 (S)
*S. maltophilia* Mc-3	16 (I)	0 (R)	0 (R)	10 (R)	0 (R)	0 (R)	10 (R)	12 (R)
*A. calcoaceticus* Mc-4	26 (S)	20 (S)	28 (S)	28 (S)	0 (R)	26 (S)	20 (S)	28 (S)
*A. calcoaceticus* MSt-6	28 (S)	26 (S)	24 (S)	28 (S)	0 (R)	24 (S)	28 (S)	24 (S)
*Pantoea* sp. MSc-1	28 (S)	20 (S)	18 (S)	22 (S)	20 (S)	25 (S)	26 (S)	18 (S)
*E. cloacae* PCt-2	12 (R)	14 (I)	10 (R)	10 (R)	0 (R)	10 (R)	18 (I)	18 (S)
*E. cloacae* Xc-3	18 (S)	0 (R)	0 (R)	16 (S)	20 (S)	10 (R)	20 (S)	22 (S)
*Pseudomonas putida* Xc-5	34 (S)	12 (R)	10 (R)	28 (S)	16 (I)	30 (S)	36 (S)	16 (I)
*Citrobacter freundii* Xc-6	18 (S)	10 (R)	0 (R)	16 (S)	14 (I)	10 (R)	14 (R)	20 (S)
*C. werkmannii* Xt-3	16 (I)	0 (R)	0 (R)	18 (S)	16 (I)	12 (R)	18 (I)	20 (S)
*S. rubidaea* Xb-3	22 (S)	12 (R)	0 (R)	18 (S)	22 (S)	14 (I)	16 (I)	22 (S)

* Letters in parenthesis indicate level of sensitivity of respective antibiotics. Affiliations: R = resistant, I = intermediate, S = sensitive. Antibiotics: AK = Amikacin; AX = Amoxicillin; CL = Cephalexin; CN = Gentamicin; NA = Nalidixic acid; TOB = Tobramycin; TE = Tetracycline; C = Chloramphenicol.

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
