# Peer review of "Functional and Genetic Diversity of Bacteria Associated with the Surfaces of Agronomic Plants"

_plants, 2019, doi:10.3390/plants8040091_

Round 1
Reviewer 1 Report
This manuscript describes the functional and genetic diversity of bacterial communities associated with agricultural produce. Authors found fresh vegetable samples colonized by several human pathogens belonging to bacterial genera Bacillus, Enterobacter, Staphylococcus, and Klebsiella. Notably, some of these bacterial strains also exhibited beneficial plant growth traits. These findings provide valuable insights into biosafety and agronomic significance of bacterial species associated with the surfaces of raw-eaten fresh vegetables such as carrot, cabbage, and turnip.
The manuscript is clearly written, and conclusions are justified. No additional experiments are requested from my side.
However, a few points need to be addressed:
1) Figures 2 and 3: How many replicates were analyzed (n=)? This should be mentioned in methods/figure legend, and appropriate error bars should be plotted to represent variation in the data set.
2) Line 175, from is missing
3) Phosphate solubilization and HCN production results are unclear to me.
First, how many bacterial isolates were evaluated for each assay? Second, provide data supporting your observations for the two assays.
4) Line 282, delete not
Author Response
I am very thankful for the worthy” reviewer 1” comments that findings of this work provided valuable insights into the biosafety and agronomic significance of bacterial species associated with the surfaces of raw-eaten vegetables. Point to point response to reviewer’s comments is given below.
1). For figures 2 and 3 that were related to auxin production and biofilm formation three replicates were placed (n = 3). It is also amended accordingly in the methodology section and in figure legends. Moreover, error bars have not been added on figure bars that represented mean of three replicates. Actually, Duncan’s multiple range test compare mean values not error bars. It is also common practice in literature to remove error bars or present date without error bars to give clear picture of comparison of data analyzed by Duncan’s multiple range test.
2). Line 175: It is actually the empty space between the table 1 and 2. It will be managed or formatted during the final setting of the manuscript.
3). Phosphate solubilization is very important microbial trait that can help plants to absorbs mineral phosphate from the soil. Similarly, HCN production by bacteria can suppress plant pathogens. These two traits were evaluated qualitatively on agar media and their significance has been elaborated in the discussion section in the revised manuscript. Data for phosphate solubilization or HCN production has been added as supplementary material.
4). Line 282: word “not” is deleted.
Reviewer 2 Report
Brief Summary
The author’s aim is to identify and evaluate the bacterial population found on three vegetable crops; carrot, cabbage and turnip. The author then evaluates this diversity based on three groups; human pathogenic bacteria, plant pathogenic bacteria and plant beneficial bacteria.
Broad comments
The author states the main aim of the study as the identification of bacterial communities associated with three agronomic crops as well as an evaluation of the diversity and agricultural significance of the results obtained. The author needs to provide a very good reason for the crops chosen for the study. One of the main issues is that while the author seems to make a case for the human pathogens found, there needs to be more work on the results for the plant pathogenic finds as well as the plant beneficial microbes. In many areas of the manuscript, there is a heavy bias toward the potential human pathogenic bacteria found. This bias needs to be corrected especially since part of the main aim is to look at the agricultural significance of the bacterial communities found. All the experiments were done at 37 degrees Celsius. This is problematic as there are many plant bacteria that grow below this temperature. By doing the experiments at 37 degrees Celsius, the author may have biased the results in favor of human pathogenic bacteria. This problem needs to be addressed.
In many sections, it is unclear how many replicates were done for the assays. There should be a minimum of three replicates per assay for reliable conclusions. The raw data obtained from the assays should also be provided in the supplementary materials. There are many areas where the bacterial names are being mentioned for the first time and the full scientific names needs to be used on those occasions. Subsequent mentions of the same bacteria can then use the abbreviated scientific names. The author need to go through the entire manuscript and rectify this problem. The author needs to provide the raw nucleotide sequences obtained with PCR. The accession numbers of the bacterial sequences already in the database (NCBI) against which the sequences of the current study were compared, also need to be provided. There are several other places in the manuscript where more information is needed: The author needs to provide pictures of the different bacterial morphologies (methods in section 2.2) found on the three crops used in the study. The author needs to provide pictures of the disc results of section 2.5. The author needs to provide pictures of the biofilm assay results from section 2.7. The author needs to provide the raw data as well as the standard curve used in section 2.6.
The discussion section of the manuscript also needs more work. In many instances, the author simply restates the results without any form of proper discussion. The author needs to talk about the implication of the work done. How do the results affect the potential for the spread of human disease? What do the results mean for the spread of plant pathogens and how can this affect policies implemented by farmers for land and crop management? How do the results affect the use of plant beneficial bacteria and how can it be used in future crop management? What do the results found mean for the potential of crop yield and economic benefits or losses in agricultural production? The author also needs to discuss why the assays are important and then talk about the implications of the assays. For example, why was the assay done with and without tryptophan? What is the importance of the phosphate solubilization assay and the hydrogen cyanide activity assay? What do these assays have to do with agronomic value? The link needs to be made in a convincing way. In the discussion section, the author needs to provide possibilities for precautions that can be taken to reduce the load of these pathogens on the food crops discussed.
The author needs to make the results applicable to other crops. How do the results fit into the context of other crop species that can be consumed raw? Also, even though the author talks about the health costs to humans, there is no mention of health costs to plants. What are the consequences of the colonization of plants by plant pathogenic bacteria as well as plant beneficial bacteria? There is also an economic cost for both humans and plants as far as the pathogenic strains are concerned and this needs to be discussed. For example, what is the cost to plant yield in the presence of pathogenic bacteria and how does this impact the farmer economically?
Specific Comments
Please see highlights in pdf:
Line 10 to 12: Please expand on this. What is the purpose of determining the genetic diversity of the bacterial community on crops? Is it to expand the scientific knowledge or for some other reason?
Line 18 to 19: What was the outcome of the screen for plant growth promoting bacteria?
Line 20 to 23: Please use the full scientific names of the bacteria mentioned here.
Line 30 to 31: Please provide citations for this statement.
Line 31 to 33: This sentence is not clear at all. Please consider changing it to "There has been a correlation between food-borne outbreaks and increased production, imports and the consumption of fresh agricultural produce".
Line 33: Please change "Raw-eaten" to "Raw".
Line 34 to 35: This statement is inaccurate. While it is true that these crops can be consumed in the raw state, there is also a lot of evidence that they can also be processed. Please moderate your language here and provide citations from food processing journals to support your claims.
Line 35 to 39: You have already stated earlier in this paragraph that there has been an increase of pathogenic microorganisms due to increased production and consumption. This section of the paragraph seems to be a repetition of what you have already said. Please delete it or consider adding new information.
Line 40 to 41: Please provide a citation for this statement.
Line 43: There are many species in the Stenotrophomonas genus. Please be clear which one you are talking about.
Line 44: Please change "Equal" to "A similar".
Line 46: This is the first time you are mentioning E.coli, please use the full name. There are also several species in the Salmonella genus. Please be clear which one you are talking about.
Line 65 to 66: Please provide the citation for this methodology.
Line 74 to 75: This sentence is misleading. According to the results, the only crops evaluated were carrot, cabbage and turnip. This sentence implies that more crops were evaluated but the results do not support that. Please modify this sentence.
Line 80 to 86: Please provide citations for these statements. Also please indicate which species of the genera mentioned you are talking about.
Line 88: Please use the plural of mortar and pestle.
Line 89 to 90: Please indicate what was in the saline solution.
Line 91 to 93: Why was the assay done at 37 degrees Celsius? There are many bacterial microbes that have optimum temperatures less than the one used in this study. The author may be biasing the results by picking this temperature. The assay needs to be repeated at lower temperatures. Please provide pictures of these different morphologies in your results section. It is fine if they are in supplemental figures. Also please describe the purification method better. A colony with a unique morphology can still be a combination of several strains and therefore it is advisable to do several rounds of purification. Describe in detail what was done for the purification.
Line 109 to 111: The MEGA software has a version 10 (MEGA-X) update. Please repeat your analysis with the latest version of the software. Also, please repeat your phylogenetic analyses with another phylogenetic method such as the maximum likelihood method. Basing your conclusions on only one method could lead to critical errors. If both methods give the same results, then you can be more confident in your conclusions. Please give details about the sequences you used for your phylogenetic analyses: the tree needs rooting. Please indicate what sequence was used for rooting the tree. Also, please provide in the supplementary material, the list of all the sequences you used to make the tree (both those you amplified and those available in the database).
Line 115 to 117: You have listed the masses of antibiotics used but this does not give any indication of the concentration of antibiotics used. Please indicate the concentrations of the antibiotics that were used. Also, another part of the study as indicated by the authors was to find plant pathogens on these crops. There are certain chemicals used on plant pathogens and the authors need to also test the resistance and susceptibility of the bacteria isolated on plant-specific antibiotics that are routinely used on plant pathogenic bacteria.
Line 124: You have previously mentioned L-agar as the abbreviation for Luria bertani agar. Since this is the first time you are mentioning Luria bertani broth, please put the abbreviation (L-broth) in brackets. Also, the designation for milliliter should be mL not ml. Please correct this throughout the manuscript.
Line 125: Which bacterial cultures did you use for the inoculation? Was it from the tissue grinding you had previously done or was it from colony cultures of the tissue grinding dilutions you had previously done. Please be very clear.
Line 126: The meaning of the centrifuge speed RPM can change from centrifuge to centrifuge. Please indicate the speed in RCF (a more consistent measure of speed) so that your experiments are more reproducible. Also indicate what type of centrifuge you used.
Line 128 to 132: Please provide the citation from which this method of detection was adapted from. Also, provide
Line 132 to 134: Please provide your version of the exact methods cited here. This can go in the supplementary material. It is unlikely that the methods cited used the exact same starting materials you used, therefore it is important that you make your adaptations of the original citations very clear. Also, assays need to be done in a minimum of three replicates. Please do that and provide the raw data in the supplementary results.
Line 137: Where did these bacterial strains come from? Were they from those purified in section 2.2? Please be clear.
Line 143: All assays need a minimum of three replicates in order to draw reliable conclusions. Please do that.
Line 153 to 155: There were several assays done: auxin production, biofilm production, phosphate solubilization, hydrogen cyanide activity. Statistical analysis needs to be done for all the assays not just for some.
Line 161: Please change "were searched" to "were used to search".
Line 162: Did you use the default settings in the BLAST search or did you modify the settings? Please be clear.
Line 163 to 168: Did all the sequences show 99% similarity or were there some which showed less? Provide the results of the BLAST results and percentages in the supplementary files. Also, a high level of similarity does not indicate absolute certainty. It is reasonable that the high similarity could indicate the genus but the authors have gone as far as to indicate the species. What other criteria are they basing the species identification on as shown in Tables 1 to 3? Please talk about the differences in the bacterial strains identified in the three crops used. Were there any species identified in one crop but not the other? Expand on the meaning of your results.
Line 170: What does "S. No." mean? If it means “Sample Number", please use the full name.
Line 179 to 188: Please use the latest version of the MEGA software. Also, the tree needs to include the other sequences already in NCBI which were used to make the determinations reported in this manuscript. Please expand on the meaning of the evolutionary relationships identified.
Line 190 to 198: Please provide information about which strains were completely absent in each crop, which strains were common in all the crops. You talk about strains specifically colonizing cabbage and carrot, what about for carrot?
Line 200 to 211: Please provide pictures of the results of the antibiotic susceptibility tests. Also, the assays need to be done in at least triplicate.
Line 213 to 226: Please provide the raw data for these assays. Also provide the pictures of the results of these assays. Also indicate which crops showed the greatest susceptibilities and resistances. Were there any crops that showed a higher resistance across all the antibiotics? Please expand on the results section. There is information to be pulled from the tables. Please summarize that information properly. Also, no figures or tables are provided for the results for the phosphate solubilization or the HCN activity. Please include these results.
Line 231 to 234: Please indicate which samples belong to which crop. Also, as I have indicated earlier, the samples already available in NCBI need to be in the tree. Provide rooting for the tree.
Line 238 to 240: Please indicate why these are in italics.
Line 258 to 265: Please summarize these results by crops. Were there more biofilm producers for one crop versus another?
Line 268 to 270: Where is the data for all the strains discovered? How were the strains shown on the graph selected? Please make this clear and have another graph in the supplementary that shows the auxin results for all the strains. The auxin graph needs error bars.
Line 274 to 276: Where is the data for all the strains discovered? How were the strains shown on the graph selected? Please make this clear and have another graph in the supplementary that shows the biofilm results for all the strains. Also, the graph needs error bars.
Line 279: Please change "are house of" to "are colonized by"
Line 280 to 283: Please provide the citations for these statements.
Line 283 to 284: Please change "studies explored" to "studies have already explored". Please summarize what these studies showed.
Line 288 to 293: You have simply restated your results. One aim of a discussion is to talk about the implications of your results and how they tie in to both previous research as well as future research. Please do that for your results.
Line 294: Delete the word "but".
Line 299 to 300: Please summarize what the study by Schoeni and Wong showed and indicate how it is linked to the results you obtained.
Line 302 to 314: It is fine that the authors talk about which the link between the bacterial species they found and other human pathogens. However, the authors need to also talk about the bacterial species they found which have been linked to both plant pathogens and as well as plant beneficial microbes. Which of their results indicate plant pathogens and which indicate beneficial microbes? What are the implications of these results?
Line 315 to 322: Please discuss the implications of your results instead of simply restating them. Also, you also need to talk about the effect of antibiotics and chemicals used on plant pathogenic bacteria, on the strains you isolated. The aim of this manuscript was to find bacteria both beneficial to plants as well as harmful to both plants and humans. The authors have focused on humans, they need to expand on the sections for plant beneficial bacteria as well as harmful plant bacteria and whether any of the isolates they found are resistant or susceptible to antibiotics or chemicals used on these three plants (carrot, cabbage and turnip) for disease prevention.
Line 323 to 334: There is a focus on the human pathogens. The author needs to talk about the auxin production by the plant pathogens as well as the plant beneficial bacteria. The author needs to discuss the results for these three groups in the context of the three crops used for the study. Are there cases where auxin production is harmful to the plant? If identified pathogenic bacteria are high auxin producers, can this be harmful to the plant? The author needs to discuss this possibility. The same approach needs to be adapted for the biofilm results. The results need to be discussed from the perspective of human pathogens, plant pathogens as well as plant beneficial bacteria. They also need to be discussed from the perspective of the three crops used in the study. Do the bacteria that specifically colonize the turnips or the cabbages show low, medium or high biofilm production? What are the implications of those results? The results for the phosphate solubilization and the HCN activity also need to be properly presented and discussed in the context of the three groups of pathogens as well as the three crops used in the study.

Author Response
I am also very thankful for the time and detailed comments of worthy “reviewer 2”. It will help to improve the text and quality of the final version of the manuscript. Revised sections of the manuscript have been highlighted in yellow color. Point to point response to reviewer’s comments follows below:
Broad comments
Indeed the main objective of this study was the identification of the bacterial communities associated with three agronomic crops or raw vegetables. Bias towards pathogenic bacteria has been addressed in the revised manuscript by adding relevant literature in the discussion section. Actually, 30-37 oC temperature range was used for experiments. In the manuscript, only highest temperature was given. It is now corrected in the methodology section. I have added 8 new references to accommodate different comments of the worthy reviewer.
For bacterial in vitro experiments, three replicates were used. During revision process of this manuscript results for biofilm or auxin production were reproduced. I have added supplementary figures or tables to support the available data. I have also added the complete nucleotide sequence file that was used for bacterial identification or phylogenetic analysis.
Discussion section has been revised keeping in view the worthy reviewer’s comments. I have also added a few more references to accommodate valuable reviewer’s suggestions. Role of L-tryptophan has been elaborated in the revised manuscript. Similarly, significance of phosphate solubilization or HCN production is also highlighted in the discussion section.
Methodological approaches that have been used in present study can be applied to other corps. I have indicated all these aspects in conclusion section.
Response to reviewer’s specific comments is given below.
Response to specific comments
1). Line 10-12: The main purpose to determine the genetic diversity of bacterial communities was to generate awareness to local community. Actually, I want to highlight that in addition to beneficial microbes plant surfaces can also be colonized or contaminated with potential human pathogens. It is now elaborated in lines 12-14 in the revised manuscript. It is also further clarified in the objective of the study in lines 72-75.
2). Outcome of the screening for plant growth prompting bacteria is elaborated with reference to their agricultural significance in lines 27-28 in the revised version.
3). Line 20-30: Full scientific names of the bacteria have been used in abstract section as per reviewer’s suggestion.
4). Line 30-31: Citation number “1” is included to support this sentence.
5). Line 33: “Raw-eaten” changed to “Raw”.
6). Line 34-35: Language of these lines have been amended in the revised manuscript.
7). Line 35-39: Sentences have been modified to avoid repetition.
8). Line 40-42: Citation number “10” is added to support this sentence.
9). Line 43: Indeed there are several species of genus Stenotrophomonas. Here, I am taking about Stenotrophomonas maltophila. It has also been revised accordingly.
10). Line 46: Full name of Escherichia coli (E. coli) is now used in first sight. Similarly, Salmonella enterica is also mentioned here as suggested by the reviewer.
11). Line 65-66: Citation number “20” is incorporated in the revised version to support the methodology.
12). Line 74-57: Sentence has been modified to accommodate the analysis of only three crops i.e. carrot, cabbage and turnip.
13). Line 80-86: Full species or scientific names are given for these sentences and supported with citation numbers “21” and “22”.
14). Line 88: Plural for “mortar and pestle” is used in the revised sentence.
15). Line 89-90: Saline solution contained 0.85% NaCl that is now given in methodology.
16). Line 91-93: Incubation temperature ranges from 30 to 37 oC that is corrected in the revised manuscript. Moreover, I have also repeated the experiment and confirmed the growth of beneficial or opportunistic pathogens at respective lower or higher temperature. Purification of strains was performed after many round of streaking. It is elaborated in the revised manuscript. I have also included the supplementary figure 1 to show the morphology of different strains and purification process by quadrant streaking.
17). Line 109-111: I tried to construct phylogenetic tree by using MEGA 10 software. But I got more consistent results with MEGA 6. To manage consistency with results I prefer to retain the previous figure. I have uploaded the available supplementary file for all the sequences.
18). Line 115-117: Concentrations of all the tested antibiotics have been mentioned in parenthesis in the revised section. Moreover, I have not worked with plant pathogenic bacteria. I have listed some potential human pathogens that have been elaborated in different sections. Therefore, specific chemicals or resistance pattern was not evaluated for isolated microbes against potential biocides.
19). Line 124: Abbreviation of Luria-Bertani broth (L-broth) is amended. Moreover, “ml” is changed to “mL” in throughout the manuscript.
20). Line 125: Purified bacterial cultures were used for different in vitro analysis. It is now mentioned in line 134 in the revised manuscript.
21). Line 126: Speed for centrifugation is now indicated in RCF along with model of the machine (Sigma 1-14) used during experimentation.
22). Line 128-134: Citation number “27” is given to support the methodology for auxin quantification. All the minor amendments are given in this section. Moreover, during revision process, I have reproduced the experiments with three replicates that are now corrected in methodology or in figure legends. I have added all the results for auxin production, phosphate solubilization or HCN production in supplementary files.
23). Line 137: Purified bacterial cultures were used for biofilm formation that is now mentioned in line 146 in the revised version of the manuscript.
24). Line 143: Experiment for biofilm formation was reproduced by using three replications. Number of replicates is corrected in the methodology.
25). Line 153-155: Statistical analysis was performed for bacterial auxin production and biofilm formation. On the other hand, phosphate solubilization and HCN production was analyzed qualitatively. That’s why statistical analysis was not included for these two traits.
26). Line 161: “were searched” changed to “were used to search”.
27). Line 162: Default settings were used for analysis that is now clarified in the revision.
28). 163-168: It’s true that bacterial species nomenclature need more experimentation. But it is mainly prefer to claim novel microbial species. I have used partial sequencing data just to indicate the presence of potential bacterial species with agronomic plants. Majority of the strains showed 99% similarity. Crop wise presence of different bacterial species is now mentioned in the section of “Bacterial diversity of fresh vegetables” lines 205-210 in the revised manuscript.
29). Line 170: S. No. means “Serial No.” that is now clarified in different tables.
30). Line 179-188: Majority of the strains gave 99% I have already given the phylogenetic tree for 55 isolated bacterial strains. This figure cannot accommodate more sequences from NCBI. Hopefully, worthy reviewer will agree with is limitation of space.
31). Line 190-198: Crop wise analysis of bacterial diversity is amended in lines 2015-210.
32). Line 200-211: Figure 2 is added to provide the results for antibiotic sensitivity pattern. Moreover, assay of antibiotics was performed in triplicate.
33). Line 213-226: Figures for auxin production, phosphate solubilization or HCN production are given in supplementary material.
34). Line 231-234: Crop wise diversity is already reported in tables 1-3. It will make figure 2 more complicated if I tried to incorporate the crop wise analysis.
35). Line 238-240: These affiliations or abbreviations are not given in italics in the revised version.
36). Line 258-265: Crop wise analysis for biofilm formation is given in lines 298-304 in the revised manuscript as suggested by the reviewer.
37). Line 268-270: In figure 3, selected strains are given here that recorded the ability to produce maximum levels of auxin in culture supernatant. Supplementary table is added to show the results of all strains.
38). Line 274-276: Significant results of biofilm formation are shown in figure 4. Full results are given in supplementary files/ tables.
39). Line 278: “are house of” changed to “are colonized by”.
40). Line 280-283: Citation number “10” is provided for this sentence.
41). Line 283-284: “studies explored” changed to “studies have already explored”. Such explanations/ information have already been given in introduction.
42). Line 288-293: These sentences that were related to results have been omitted/ deleted from discussion as per reviewer’s suggestion.
43). Line 294: Word “but” is deleted and sentence has been rephrased.
44). Line 299-300: Study of Schoeni and Wong have been summarized and link is established with present study.
45). Line 302-314: Implications of potential human pathogens have been included in discussion in the revised manuscript with examples of bacterial species.
46). Line 315-322: Discussion section has been revised keening in view the comments of reviewer.
47). Line 323-334: Reference number 20 and 46-49 have been added in the discussion section to high light the benefits of bacteria exhibiting different beneficial traits. I have specifically focused on potential human pathogens that may exhibit beneficial plant traits, especially, auxin production, phosphate solubilization, biofilm formation and HCN production.
Round 2
Reviewer 2 Report
Brief Summary
The author’s aim is to identify and evaluate the bacterial population found on three vegetable crops; carrot, cabbage and turnip. The author then evaluates this diversity based on three groups; human pathogenic bacteria, plant pathogenic bacteria and plant beneficial bacteria. The objective is to evaluate the potential biosafety and agronomic significance of the microbe populations identified.
Broad comments
The author states the main aim of the study as the identification of bacterial communities associated with three agronomic crops as well as an evaluation of the diversity and agricultural significance of the results obtained. The author has made some improvements in the revised manuscript but more work is needed. In some areas of the manuscript, there is still some bias toward either plant beneficial microbes or toward human pathogens. For example, the author only has a few sentences talking about the phosphate solubilization and HCN production results. This section needs to be expanded in both the results and the discussion for both the potential human pathogens as well as the potential plant beneficial microbes. The author has also attempted to clarify the temperatures at which experiments were conducted but this needs to be made clearer. In the response to reviewers, the author states that the experiments were done between 30 to 37 degrees Celsius. However in many places, the author only mentions 37 degrees Celsius. The author needs to be very clear here because if the bacteria were isolated at 30 degrees, the subsequent experiments also should have been conducted at 30 degrees to ensure optimum results. The author has also attempted to clarify the number of replicates done for the experiments and this is commendable. However, since the experiments were done in triplicate, it is therefore possible to calculate the standard error of the mean (SEM) and this needs to be done and error bars need to be included in all of the graphs. The author also needs to include more information in the figure and table captions and legends. Some of the figures have very poor resolution and this needs to be corrected. The author includes supplementary material but none of this supplementary material is referenced in the main part of the manuscript and this needs to be corrected. The author needs to provide pictures of the biofilm assay results from section 2.7. The author needs to provide the raw data as well as the standard curve used in section 2.6. A lot of the pictures and tables that need to be in the results section are in the discussion section and this needs to be corrected.
The discussion section of the revised manuscript also needs more work. None of these questions from the first round of review was adequately answered: How do the results affect the potential for the spread of human disease? What do the results mean for the spread of plant pathogens and how can this affect policies implemented by farmers for land and crop management? How do the results affect the use of plant beneficial bacteria and how can it be used in future crop management? What do the results found mean for the potential of crop yield and economic benefits or losses in agricultural production? The author also needs to discuss why the assays are important and then talk about the implications of the assays. For example, why was the assay done with and without tryptophan? What was the purpose of artificially inducing auxin production? What are the differences between the auxin production with and without tryptophan and why was it necessary to cause an artificial increase in the auxin levels? What is the importance of the phosphate solubilization assay and the hydrogen cyanide activity assay? What do these assays have to do with agronomic value? The link needs to be made in a convincing way. In the discussion section, the author needs to provide possibilities for precautions that can be taken to reduce the load of these pathogens on the food crops discussed.
The author needs to make the results applicable to other crops. How do the results fit into the context of other crop species that can be consumed raw? Also, even though the author talks about the health costs to humans, there is no mention of health costs to plants. What are the consequences of the colonization of plants by plant pathogenic bacteria as well as plant beneficial bacteria? There is also an economic cost for both humans and plants as far as the pathogenic strains are concerned and this needs to be discussed. For example, what is the cost to plant yield in the presence of pathogenic bacteria and how does this impact the farmer economically? The answers to these questions are important because the author lists part of the main objective as evaluating the agronomic significance of the microbes identified.
Specific Comments
Please see green highlights in pdf:
Line 11: Please change this to "selected agronomic plants".
Line 19 to 20: Please use the concentrations (mass per volume) of these antibiotics instead of the masses. The masses tell the reader nothing about reproducing the effective concentrations.
Line 21 to 23: What do the high auxin and biofilm productions mean? You need to summarize the significance of your findings in the abstract instead of simply stating your findings.
Line 33: Please change this to "components of the human diet".
Line 122: Was this the purified strains or the initial isolation? Please be clear.
Line 124: Please change this to "antibiotic susceptibility test discs".
Line 128: If this was done at a temperature range as indicated in the reply to the reviewer, please indicate that here.
Line 136: Please be clear about your temperature ranges.
Line 149: Please be clear about your temperature ranges.
Line 155: Please be clear about your temperature ranges.
Line 178: Please provide more information in your table caption. What media and what temperatures were used in the isolation? Since the author indicates that a temperature range was used, it needs to be indicated which strains were isolated with which temperatures.
Line 180: Please provide more information in your table caption. What media and what temperatures were used in the isolation? Since the author indicates that a temperature range was used, it needs to be indicated which strains were isolated with which temperatures.
Line 182: Please provide more information in your table caption. What media and what temperatures were used in the isolation? Since the author indicates that a temperature range was used, it needs to be indicated which strains were isolated with which temperatures.
Line 187: What sort of relationship are you talking about? If you mean the evolutionary relationship, please say so.
Line 262 to 263: Please provide more information in your figure caption. What media and what temperatures were used in the isolation? Since the author indicates that a temperature range was used, it needs to be indicated which strains were isolated with which temperatures. The author also needs to indicate that the tree was made with trimmed nucleotide sequences obtained after amplification of the 16s gDNA.
Line 285 to 288: Where are the figure references for these statements and observations?
Line 290 to 292: Please indicate if these were the purified bacterial isolates. Also indicate what temperature the 72 hour incubation was done at.
Line 297: Figure 3 is the auxin production. I think you mean figure 4. Also, you have already stated that OD greater than one is the benchmark for good biofilm production (line 292 to 293), you have repeated yourself here, please correct this. Also, for the first place where you introduce optical density, indicate the abbreviation in brackets and then from then onwards, use the abbreviation.
Line 309: Please change "presenting" to "present".
Line 315: please delete "however" and remove the full stop punctuation and add a comma.
Line 330 to 331: Please summarize the results of this study.
Line 335: Please change this to "produce-related".
Line 337: The sentence here is incomplete. The author needs to correct this. Also, please provide more information in your figure caption. What media and what temperatures were used in the isolation? Since the author indicates that a temperature range was used, it needs to be indicated which strains were isolated with which temperatures. The authors also need to indicate how long the antibiotic susceptibility discs were left on the plates and at what temperature the assays were done. The author also needs to indicate that the results were obtained with purified isolates. Also the resolution of the picture is extremely poor. Please provide new pictures. Also, the picture belongs in the results section, not in the discussion.
Line 340: Please provide more information in your table caption. What media and what temperatures were used in the isolation? Since the author indicates that a temperature range was used, it needs to be indicated which strains were isolated with which temperatures. The authors also need to indicate what millimeter range zone around the discs equal susceptibility or resistance. For example, "range x to x mm indicates susceptibility while range x to x mm indicates resistance". The author also needs to indicate how many times, this antibiotic study was done. The author also needs to indicate that the results were obtained with purified isolates. Also, the table belongs in the results section, not in the discussion.
Line 351 to 352: Please provide more information in your table caption. What media and what temperatures were used in the isolation? Since the author indicates that a temperature range was used, it needs to be indicated which strains were isolated with which temperatures. The authors also need to indicate what millimeter range zone around the discs equal susceptibility or resistance. For example, "range x to x mm indicates susceptibility while range x to x mm indicates resistance". The author also needs to indicate how many times, this antibiotic study was done. The author also needs to indicate that the results were obtained with purified isolates. Also, the table belongs in the results section, not in the discussion.
Line 362 to 365: Please provide more information in your figure caption. What media and what temperatures were used in the isolation? Since the author indicates that a temperature range was used, it needs to be indicated which strains were isolated with which temperatures. The author also needs to indicate that the results were obtained with purified isolates. Also, since the results here are from a mean of three replicates, this is enough to calculate a standard error of the mean (SEM) and therefore error bars are needed. Please provide the error bars. Also this figure belongs in the results section, not the discussion.
Line 369 to 371: Please provide more information in your figure caption. What media and what temperatures were used in the isolation? Since the author indicates that a temperature range was used, it needs to be indicated which strains were isolated with which temperatures. The author also needs to indicate that the results were obtained with purified isolates. The results here are a mean of three replicates and therefore there is enough information for the standard error of the mean (SEM), error bars are therefore needed. Please calculate the SEM and provide the error bars. Also, this figure belongs in the results section, not in the discussion.
Line 374: Please delete "therefore"
Line 375 to 379: Please provide the figure or table references for these statements.
Line 383 to 384: Are you talking about the current study or previous studies? Please be clear.
Line 387: Please change this to "the control".
Line 388: The author needs to compare the results before and after tryptophan addition. What is the level of production in these microbes without the addition of the amino acid and why was it necessary to add and artificially increase the level of auxin production? What was the author trying to achieve by this artificial production? Under normal circumstances, plants could not be exposed to this level of tryptophan, therefore the author needs to justify this result and explain why it is relevant.
Line 389 to 391: Please expand on the biofilm section for the potential human pathogens. Which strains that are potential human pathogens were high biofilm producers? Were any biofilm producers specific to any of the three crops investigated? What are the implications of consuming potential human pathogenic strains that are biofilm producers?
Line 400 to 402: Please expand on the beneficial ways that these bacteria affect their hosts.
Line 403 to 404: Please expand on the benefits of phosphate solubilization. What is the role of phosphate for plant growth? What are the causes and consequences of phosphate deficiency? Does it affect yield, quality, immunity? Please link the results you got to these advantages or disadvantages of phosphate solubilization. Were there any strains specific to any of the three crops that had higher level of solubilization compared to others? What are the implications of your results? Were there any strains specific to any of the crops investigated that produced more HCN than others? Expand on how HCN suppresses phytopathogens and then talk about the implications of the results you obtained on the three crops investigated. Which of the strains that solubilize phosphate and/or produce HCN are potential human pathogens and what are the potential benefits or harmful effects to humans when they consume these raw vegetables that have strains that solubilize phosphate and produce HCN?
Line 407: Delete "but".
Line 409: Please change this to "may be difficult to completely remove"
Supplementary File
Supplementary file with nucleotide sequences. Please give this a name and reference it in the main body of the manuscript text.
Figure S1: Please provide more information in your figure caption. What temperatures were used in the isolation? Since the author indicates that a temperature range was used, it needs to be indicated which strains were isolated with which temperatures. Also, the resolution of the picture is very poor, please provide new pictures. This picture also needs to be referenced in the main body of the manuscript.
Figure S2: Please provide more information in your figure caption. What media and what temperatures were used in the isolation? Since the author indicates that a temperature range was used, it needs to be indicated which strains were isolated with which temperatures. The author also needs to indicate that the results were obtained with purified isolates. The author needs to indicate what qualitative measure was used to judge the production of HCN. For example "the dark orange color indicates X while the light orange color indicates X". It is impossible to tell from the figure which are a, b, c, d, e and so on. Please correct this. This figure also needs to be referenced in the main body of the manuscript.
Figure S3: The strain name needs to be italicized. Please provide more information in your figure caption. What temperatures were used in the isolation? Since the author indicates that a temperature range was used, it needs to be indicated what particular temperature was used for the strain in the figure. The author also needs to indicate that the results were obtained with purified isolates. The author also needs to indicate what qualitative measure is being used to judge phosphate solubilization. A figure of phosphate solubilization vrs non solubilization is needed to make this figure relevant and impactful. This figure also needs to be referenced in the main body of the manuscript.
Table S1: Please provide more information in your table caption. What media and what temperatures were used in the isolation? Since the author indicates that a temperature range was used, it needs to be indicated which strains were isolated with which temperatures. The author also needs to indicate how many times the assays were done and since done in triplicate, provide the standard error of the mean calculation results next to each mean assay result. The author also needs to indicate that the results were obtained with purified isolates. This table also needs to be referenced in the main body of the manuscript.
Table S2: Please provide more information in your table caption. What media and what temperatures were used in the isolation? Since the author indicates that a temperature range was used, it needs to be indicated which strains were isolated with which temperatures. The author also needs to indicate that the results were obtained with purified isolates. This table needs to be referenced in the main body of the manuscript.

Author Response
RESPONSE TO REVIEWER COMMENTS
I am very thankful for the effort, time and detailed comments of worthy reviewer. It will help to improve the text and quality of the final version of the manuscript. Revised sections of the manuscript have been highlighted in yellow color. Point to point response to reviewer’s comments follows below:
Broad comments
The major concern of worthy reviewer about the range of temperatures used for isolation has been addressed properly in the revised manuscript. I have added two columns in tables 1-3 that are related to “Temperature of isolation” and “Culture media”. I have tried to update the tables according the available information (for temperature and media). For Phosphate solubilization and HCN production more information has been added in discussion section and supported with two additional references. In figure 3 and 4, standard errors have been added as per reviewer’s suggestion. Figures and tables given in supplementary files have been referenced in manuscript text. For biofilm assay, I used 96 wells plate to record optical density for different shades of crystal violet dye and then plates were discarded. Such record usually not maintained for photography. Moreover, for auxin production, I have added the range of the standard auxin concentrations processed to construct standard curve. Actually, I draw standard curve manually on graphic paper that I think may not be suitable as supplementary file.
Discussion section has been revised by adding 3 more references. Bacterial concerns related to the spread of diseases have discussed in the revised manuscript. Moreover, control of fresh produce related outbreaks have also been suggested by managing pre or postharvest practices. Significance of different biochemical assays has been justified in discussion section. Bacterial screening for different plant growth promoting traits may have agricultural application as biofertilizers to enhance crop productivity. L-tryptophan dependent auxin production has also been discussed in the revised manuscript.
Our main focus of research was to isolate general bacterial diversity and then it’s classification into plant beneficial or opportunistic human pathogens. Actually, we have not targeted the isolation of plant pathogens. Moreover, we have not investigated the effect of bacterial colonization on yield parameters or farmer economy. Bacterial agronomic significance was evaluated in terms of microbial beneficial traits that can mediate plant growth and development. I have not focused on agronomic significance with reference to crop or farmer economy. Undoubtedly, reviewer’s suggestions about these issues are genuine and can be considered as a separate research topic. In conclusion, I have suggested that methodological strategies used in current work can be used for the investigation or screening of general or specific bacterial diversity associated with other crops of fresh produce. Response to reviewer’s specific comments is given below.
Response to specific comments
Line 11: Phrase changed to “selected agronomic plants”
Line 10-20: Concentrations of different antibiotics are mentioned in µg/ disc. It is fixed quantity that is mentioned by manufacturers for each antibiotic disc. We have not used/ dissolved antibiotics in liquid. It is the standard protocol that is according to manufacturer’s instruction and followed according reference number [26].
Line 21-23: Significance of auxin production and biofilm formation is mentioned in the concluding section of abstract.
Line 33: Phase changed to “components of the human diet”.
Line 122: Purified cultures of bacteria were used that is now clarified in section 2.5.
Line 124: Phrase changed to “antibiotic susceptibility test discs” as per reviewer’s suggestion.
Line 128: Antibiotic assay was performed at 37 oC for 24 h. It is according to the protocol of Bauer et al. that is supported with reference number [26].
Line 136: Auxin production was evaluated at 37 oC for 24 h. Microbes produce high quantities of auxin at this temperature.
Line 149: Biofilm formation was recorded at 37 oC.
Line 155: Assay was performed at 37 oC.
Line 178: In table 1, two columns have been incorporated to add information related to “Temperature for isolation” and “Culture media” for different bacterial strains. According to available information, I have mentioned respective general or selective media with isolation temperature. Hopefully, this revision will help to remove the confusion related to range of temperature or culture media used during the isolation process.
Line 180: In table 2, two columns have been incorporated to add information related to “Temperature for isolation” and “Culture media” for different bacterial strains. According to available information, I have mentioned respective general or selective media with isolation temperature. Hopefully, this revision will help to remove the confusion related to range of temperature or culture media used during the isolation process.
Line 182: In table 3, two columns have been incorporated to add information related to “Temperature for isolation” and “Culture media” for different bacterial strains. According to available information, I have mentioned respective general or selective media with isolation temperature. Hopefully, this revision will help to remove the confusion related to range of temperature or culture media used during the isolation process.
Line 187: Phylogenetic tree showed an evolutionary relationship that is now mentioned in the revised article.
Line 262-263: In tables 1-3, I have mentioned “culture media” and “temperature for isolation” for each strain. I think this information is sufficient to remove confusion in different sections of the manuscript. In figure 1, I have mentioned that nucleotide sequences were trimmed after 16S rRNA gene sequencing to construct phylogenetic tree.
Line 285-288: Figure S2 and figure S3 now referenced for these results.
Line 290-292: Purified bacterial cultures were used for all assays. Temperature is given according to reviewer’s suggestion.
Line 297: Figure number of biofilm assay is corrected. Repetition of sentence is corrected and optical density (OD) is abbreviated in first sight.
Line 309: “Presenting” change to “present”.
Line 315: “However” is deleted and comma is added.
Line 330-331: Results related to staphylococcal enterotoxins (SEs) are summarized in lines 354-356 in the revised manuscript.
Line 335: Phrase changed to “produce-related”
Line 337: Typographic or setting error for figure 2 caption is corrected. In tables 1-3, I have mentioned “culture media” and “temperature for isolation” for each strain. I think this information is sufficient to remove confusion in different sections of the manuscript. Assay was performed with purified bacterial cultures and plates were incubated at 37 oC for 24 h. It is now corrected in figure 2 legend. Figure 2 is now placed in the result section.
Line 340 and 351-352: In tables 1-3, I have mentioned “culture media” and “temperature for isolation” for each strain. I think this information is sufficient to remove confusion in different sections of the manuscript. Assay was performed with purified bacterial cultures by using three set of plates for each strain or antibiotic and plates were incubated at 37 oC for 24 h. Table 4 and 5 titles are amended as per reviewer’s suggestion. Zones were measured in millimeter as per instruction of the manufacturer (Bioanalyse®). Moreover, zones were compared with the standardized chart for antibiotics (M100-S23) given by clinical laboratory standard institute (CLSI, 2013). The addition of zone range for each antibiotic will make the table very complicated. Generally, above mentioned practice or reference is used for explanation. Table 4 and 5 are now correctly placed before discussion.
Line 362-365: In tables 1-3, I have mentioned “culture media” and “temperature for isolation” for each strain. I think this information is sufficient to remove confusion in different sections of the manuscript. We used purified cultures for all biochemical tests. I have added the error bars in figure 3. Figure caption has been modified accordingly. Figure 3 is placed in result section.
Line 369-371: In tables 1-3, I have mentioned “culture media” and “temperature for isolation” for each strain. I think this information is sufficient to remove confusion in different sections of the manuscript. We used purified cultures for all biochemical tests. I have added the error bars in figure 4. Figure caption has been modified accordingly. Figure 4 is placed before discussion section.
Line 374: “therefore” is deleted.
Line 375-379: Reference of table 4 and 5 is quoted for these statements.
Line 383-384: I am talking about current study. It is now corrected.
Line 387: “control” is changed to “the control”.
Line 388: Comparison of results with and without L-tryptophan is given in lines 376-383 in the discussion section. I have demonstrated the artificial production of auxin to show the bacterial ability to utilize precursor (L-tryptophan) and convert it to auxin. In nature, microbes convert naturally available L-tryptophan in root exudates or soil to auxin that may be absorbed by the plant during early stages of development. This statement is also supported with reference number [45].
Line 389-391: Biofilm section is expanded with reference to potential human pathogens or their implications in lines 386-390. Moreover,
Line 400-404: Benefits of phosphate solubilization or its deficiency have been discussion and supported with reference number [52]. Phosphate or HCN production was recorded qualitatively and most efficient strains are mentioned in Table S2 and S3. HCN production by a few opportunistic pathogens is also discussed.
Line 407: “but” is deleted.
Line 409: Phrase is changed to “may be difficult to completely remove”.
Response to supplementary file
1). Reference of supplementary file with nucleotide sequence is mentioned in section 3.1.
2). Figure S1 is now referenced in the text. I have already provided the information related to temperature and culture media in tables 1-3. I have used the best available figures. I have tried to enhance the resolution by changing settings or adjustments.
3). Figure S2 is now referenced in the text. I have already provided the information related to temperature and culture media in tables 1-3. I have already used the best available figures. I have tried to enhance the resolution by changing settings or adjustments. For HCN production, intensity of the color indicates more positive results. These results were performed qualitatively with purified bacterial strains. Figure caption is also corrected.
4). In figure S3, strain name is italicized. I have already provided the information related to temperature and culture media in tables 1-3. I have already used the best available figures. I have tried to enhance the resolution by changing settings or adjustments. Moreover, I have photographed only those plates that showed relatively good results or clear zones. Phosphate solubilizatin assay was performed qualitatively. More clear zones indicate high phosphate solubilization ability by the bacterial strains. Figure S3 is also referenced in the text.
5). Table S1 has been referenced in the text. I have already provided the information related to temperature and culture media in tables 1-3. Assay for auxin production and biofilm formation was performed in triplicate. Mean values of these assays were compared by using Duncan’s multiple range test. This test compare only mean values of different treatment not standard errors. Moreover, addition of standard errors will make the table more complicated. In literature, mean values have also been compared without error bars. Nevertheless, I have added error bars in figure 3 and 4 as per reviewer’s suggestion.
6). For Table S2, we got results by using purified bacterial cultures. Table reference is also given in text. I have already provided the information related to temperature and culture media in tables 1-3.
Round 3
Reviewer 2 Report
Brief Summary
The author’s aim is to identify and evaluate the bacterial population found on three vegetable crops; carrot, cabbage and turnip. The author then evaluates this diversity based on three groups; human pathogenic bacteria, plant pathogenic bacteria and plant beneficial bacteria.
Broad comments
The author has made changes to the manuscript that highlight the work done better and add value to the significance of the study. In general there are several instances in the manuscript where the author uses “µl” instead of “µL” and a few cases of “ml” instead of “mL”. This needs to be corrected. The picture quality remains poor and this needs to be addressed.
Specific Comments
Please see green highlights in pdf:
Line 61: Please change "persistence" to "survival".
Line 92: The "l" in microliter should be capitalized to "L". Please check the entire manuscript and ensure that any such instances are corrected.
Line 134: The "l" in milliliter should be capitalized to "L". Please check the manuscript and correct all such instances.
Line 208: Delete the word "whereas".
Line 304: Please change "biolim" to "biofilm".
Line 305: Please change "recorded" to "showed".
Line 308: Please change this to "to be very effective as a biofilm producer".
Line 344: Please change "that included" to "known as".
Line 366: Please change "record" to "show".
Line 392: Please change "food chain" to "the food chain".
Line 394: Please change "IAA" to "IAA production".

Author Response
Response to reviewer's 2 comments
I am very thankful for the effort and time of the worthy reviewer to indicate minor setting or spelling errors during the third round of revision. Modified sections of the manuscript have been highlighted in yellow color. Point to point response to reviewer’s specific comments follows below:
Line 61: “Persistence” changed to “survival”
Line 92: The “l” for microliter has been changed to capitalized letter “L” throughout the manuscript.
Line 134: The “l” for milliliter has been changed to capitalized letter “L” throughout the manuscript.
Line 208: “Whereas” is deleted in the revised manuscript.
Line 304: “biolim” changed to “biofilm”
Line 305: “recorded” changed to “showed”.
Line 308: Phrase is changed to “to be very effective as a biofilm producer”
Line 344: “that included” changed to “known as”
Line 366: “record” changed to “show”.
Line 392: “food chain” changed to “the food chain”
Line 394: “IAA” changed to “IAA production”